# Targeted Treatment against Lipoprotein (a): The Coming Breakthrough in Lipid Lowering Therapy

**DOI:** 10.3390/ph15121573

**Published:** 2022-12-16

**Authors:** Bożena Sosnowska, Stanisław Surma, Maciej Banach

**Affiliations:** 1Department of Preventive Cardiology and Lipidology, Medical University of Lodz, 93-338 Lodz, Poland; 2Faculty of Medical Sciences in Katowice, Medical University of Silesia, 40-752 Katowice, Poland; 3Cardiovascular Research Centre, University of Zielona Gora, 65-417 Zielona Gora, Poland; 4Department of Cardiology and Adult Congenital Heart Diseases, Polish Mother’s Memorial Hospital Research Institute (PMMHRI), 93-338 Lodz, Poland

**Keywords:** atherosclerotic cardiovascular diseases, lipoprotein (a), pelacarsen, olpasiran, SLN360

## Abstract

Atherosclerotic cardiovascular diseases (ASCVD) are a very important cause of premature death. The most important risk factor for ASCVD is lipid disorders. The incidence of lipid disorders and ASCVD is constantly increasing, which means that new methods of prevention and treatment of these diseases are still being searched for. In the management of patients with lipid disorders, the primary goal of therapy is to lower the serum LDL-C concentration. Despite the available effective lipid-lowering therapies, the risk of ASCVD is still increased in some patients. A high level of serum lipoprotein (a) (Lp(a)) is a risk factor for ASCVD independent of serum LDL-C concentration. About 20% of Europeans have elevated serum Lp(a) levels, requiring treatment to reduce serum Lp(a) concentrations in addition to LDL-C. Currently available lipid lowering drugs do not sufficiently reduce serum Lp(a) levels. Hence, drugs based on RNA technology, such as pelacarsen, olpasiran, SLN360 and LY3819469, are undergoing clinical trials. These drugs are very effective in lowering the serum Lp(a) concentration and have a satisfactory safety profile, which means that in the near future they will fill an important gap in the armamentarium of lipid-lowering drugs.

## 1. Background

### 1.1. Atherosclerotic Cardiovascular Diseases

The prevalence of cardiovascular disease (CVD) has steadily increased over the years. In 2019, 523 million people worldwide suffered from CVD (in 1990 the figure was 271 million), while the number of deaths from CVD was 18.6 million (in 1990 the figure was 12.1 million) [1]. Today, all over the world, CVD is the leading cause of premature death. The vast majority of CVDs are atherosclerotic cardiovascular diseases (ASCVD), including coronary artery disease (CAD), stroke and peripheral artery disease (PAD) [2].

For years, lipid disorders have been in the third place among the most common ASCVD risk factors in the world [1]. The assessment of the lipid profile plays a fundamental role in the analysis of ASCVD risk [3]. In primary cardiovascular prevention, risk assessment is based on the SCORE (systemic coronary risk estimation) or newer scores—SCORE2 and SCORE2-OP (older persons) scales [3,4,5]. The SCORE scale assesses the total cholesterol serum concentration, while the SCORE2 and SCORE2-OP scales include the non-HDL cholesterol serum concentrations [4].

The goals of lipid-lowering treatment are determined on the basis of the risk group to which the patient has been assigned. The primary goal of lipid-lowering treatment is to adequately reduce the serum concentration of low-density lipoprotein cholesterol (LDL-C) [5]. Lipid-lowering treatment is very effective, both in primary and secondary cardiovascular prevention [6,7]. Lipid-lowering therapy is the most effective cardiac therapy to prevent ASCVD. After 5 years, there was a reduction in risk of about 20–25%, and after 40 years, even 50–55% for each mmol/L of LDL-C [8]. However, it should not be forgotten that despite the effective reduction of LDL-C, in some patients there remains a very significant influence on the cardiovascular risk of residual factors, which in the case of lipidology include non-HDL cholesterol, triglycerides, and lipoprotein (a). Serum non-HDL cholesterol is the secondary goal of lipid-lowering therapy [5].

Serum lipoprotein (a) (Lp(a)) concentration is a very important, but not fully appreciated, a risk factor for ASCVD [5]. Increased serum Lp(a) concentration is an important risk factor for ASCVD because it is independent of the serum LDL-C [9]. Until now, targeted therapies for Lp(a) have not been available. Currently, a number of clinical trials are conducted with the use of drugs that significantly reduce the serum Lp(a) concentration.

### 1.2. Lipoprotein (a)—Brief Overview for Clinicians

The Lp(a) molecule, structurally similar to the LDL particle, contains apolipoprotein (a) (apo (a)) which is conventionally linked to ApoB_100_ via a disulfide bonds [10]. Lp(a) serum concentrations are genetically determined (90%); lifestyle changes have no effect on these concentrations [10]. So far, the correct values of serum Lp(a) concentration have not been established. It is indicated that the serum Lp(a) concentration, both fasted and fed, should be lower than 30 mg/dL (78 nmol/L) [5]. Epidemiological studies have shown that 20% of subjects have a genetically determined increased serum Lp(a) concentration (>50 mg/dL [125 nmol/L]) (Figure 1) [11].

Lp(a) is an independent CVD risk factor, as evidenced by the fact that up to 30% of patients with familial hypercholesterolaemia and/or acute coronary syndrome may have elevated levels of this lipoprotein at the desired LDL-C levels [14]. Based on the serum Lp(a) concentration, the cardiovascular risk can be determined: 30–50 mg/dL (75–125 nmol/L) moderate risk; >50 mg/dL (125 nmol/L) high risk; >180 mg/dL (450 nmol/L) very high cardiovascular risk [5]. Despite this, the latest EAS guidelines still suggest a cut-off point for Lp(a) >50 nmol/L [15]. Please note that according to all recommendations, we should strive to determine Lp(a) in nmol/L, because this determines not the mass of particles, but their number [5,15]. 

In a study by Rikhi et al., including 4585 subjects from Multi-Ethnic Study of Atherosclerosis (MESA), with 13.4 years of follow-up, it was shown that when Lp(a) was elevated, the risk of ASCVD events increased, regardless of baseline LDL-C [9]. A study by Afshar et al. involving 2606 Framingham Offspring participants assessed the correlation between serum Lp(a) and LDL-C concentrations and the risk of ASCVD. It was also found that elevated serum Lp(a) concentrations, regardless of LDL-C, were associated with an increased risk of ASCVD [16]. In a study by Kaiser et al., involving 191 patients with ASCVD, it was found that high serum Lp(a) concentrations were associated with accelerated progression of low-attenuation plaque (necrotic core) in patients with advanced multivessel coronary artery disease, despite receiving guideline- based preventative therapies [17]. Moreover, determination of serum Lp(a) concentrations may contribute to a more accurate risk assessment of ASCVD. In a study by Nurmohamed et al., involving 12,437 subjects, it was shown that those with serum Lp(a) > 99th percentile had an OR of 2.64 for ASCVD [95% CI: 1.45–4.89] and 3.39 for MI (95% CI: 1.56–7.94). Importantly, the addition of Lp(a) to ASCVD risk algorithms led to 31% and 63% being reclassified into a higher risk category for SCORE and Second Manifestations of ARTerial disease (SMART), respectively [18]. In a study by Willeit et al. involving 826 subjects, it was found that elevated Lp(a) serum concentrations predicted 15-year ASCVD outcomes and improved ASCVD risk prediction, particularly in intermediate-risk groups [19]. A study by Kamstrup et al. involving 8720 subjects found that extreme Lp(a) serum concentrations substantially improved MI and CHD risk prediction [20].

Further evidence for a significant influence of Lp(a) on the risk of ASCVD was provided by the results of clinical trials in which the effect of lowering the concentration of this lipoprotein on the cardiovascular risk was assessed. In the FOURIER study, in a group of patients with stable coronary artery disease treated with evolocumab, a reduction in serum Lp(a) concentration by 26.9% (6.2–46.7%) and a reduction of cardiovascular incidents by 23% (HR = 0.77; 95% Cl: 0.67–0.88) was achieved in subjects with baseline Lp(a) above the median (37 nmol/L; 15 mg/dL), while in the group with Lp(a) below the median only by 7% (HR = 0,93; 95% Cl: 0.80–1.08). NNT (Number Needed-to-Treat) was 41 and 105, respectively [21]. There was a significant correlation between a 15% reduction in the risk of major coronary events (95% Cl: 2–25%; *p* = 0.0199) and a reduction in serum Lp(a) concentration by 25 nmol/L after adjustment for LDL-C [21]. A sub-analysis of the ODYSSEY OUTCOMES study in ACS patients treated with alirocumab showed similar results. Risk reduction after 4 months of treatment analyzed in patient groups with baseline Lp(a) < 6.7 mg/dL, 6.7 to 21.2 mg/dL, 21.2 to 59.6 mg/dL and ≥59, 6 mg/dL was, respectively, 5% (HR = 0.95; 95% Cl: 0.97–1.15), 15% (HR = 0.85; 95% CI: 0.71–1.03), 21% (HR = 0.79; 95% CI: 0.66–0.94) and 17% (HR = 0.83; 95% CI: 0.70–0.98). A 5 mg/dL reduction in Lp(a) was associated with a significant 2.5% reduction in cardiovascular events [22,23]. A reduction in the risk of coronary heart disease has also been shown in an analysis of 62,240 patients with coronary heart disease compared with a control group of 127,000 patients. It was shown that each 10 mg/dL reduction in Lp(a) was associated with a 5.8% reduction in the risk of coronary heart disease (OR = 0.94; 95% Cl: 0.93–0.95). In turn, a reduction in LDL-C concentration by 10 mg/dL resulted in a significant reduction of this disease by 14.5% (OR = 0.86; 95% Cl: 0.82–0.89). It has been shown that decreasing Lp(a) levels by 101.5 mg/dL achieved a similar reduction in ischemic disease as reducing LDL-C by 38.7 mg/dL [24].

Table 1 summarizes the most important results of the above studies.

Currently available lipid-lowering treatments insufficiently protect patients against high serum Lp(a) concentration (Table 2) [5].

It should also be mentioned that lifestyle changes have almost no effect on serum Lp(a) concentrations [27,28].

Considering the numerous data indicating a significant role of Lp(a) in increasing the risk of ASCVD, the guidelines of the Polish Lipid Association (PLA) indicate in which patients should be measured the concentration of this lipoprotein (Table 3) [5].

Overall, Lp(a) is an important and independent risk factor for ASCVD that is elevated in a significant proportion of the population. Currently available lipid-lowering therapies do not adequately control serum Lp(a) concentrations, hence new drugs based on RNA technology are under investigation to target Lp(a).

### 1.3. Lp(a) and ASCVD—Brief Overview of Pathophysiological Mechanisms

The influence of Lp(a) on the risk of ASCVD is related to its atherosclerotic, pro-inflammatory and pro-thrombotic effects [11,29].

The pro-inflammatory effects of Lp(a) are associated with an increase in macrophage IL-8 expression (oxidized phospholipids (ox-PLs). Modification of apo(a) induces IL-8 expression), monocyte cytokine release, oxidized phospholipids, and monocyte chemotaxis/transmigration (carried by Lp(a) as chemotactic factors, induces monocyte activation and migration in the sub-endothelial space), and carries MCP-1 (ox-PLs contained in Lp(a) bind and induce the production of MCP-1, a chemokine that mediate initiation and progression of vascular inflammation). Moreover, Lp(a) increases the production of adhesion molecules such as vascular cell adhesion molecule 1 (VCAM-1) and E-selectin [11,29].

In turn, the pro-atherosclerotic effect of Lp(a) results from the increase of endothelial cell growth and binding, upregulation of adhesion molecules promoting proliferation and migration vascular smooth muscle cell (VMSC) in atherosclerotic lesions (by Lp(a), apo(a) and ox-PLs), proteoglycan matrix binding, foam cell formation (ox-PLs contained in Lp(a) by inducing uptake by monocyte-macrophages leading to foam cell formation), necrotic core formation, and lesion calcification. Ox-PLs contained in Lp(a) induce reactive oxygen species (ROS) generation, and thereby cell apoptosis, which contributes to atherosclerotic plaque rapture. Lp (a), due to its small diameter (<70 mm), can flow freely through the endothelial barrier and, like the LDL particle, be retained in the artery wall and directly contribute to the progression of atherosclerotic lesions. Lp(a) enters to intima without receptors mediators, but by a mechanism dependent on lipoprotein plasma concentrations [5,11,29].

Lp(a) is also characterized by strong prothrombotic properties. Apo(a) completes with plasminogen for fibrin affinity sites, and small apo(a) isoforms have a higher affinity than large isoforms. Apo(a) promotes platelet aggregation and degranulation via the thrombin receptor. Lp(a) increases tissue factor (TF) expressions and inactivates TF pathway inhibitors which finally promotes thrombosis. Lp(a), by inhibiting the binding of plasminogen to endothelial cell surface receptors, interferes with plasmin synthesis and fibrinolysis. Moreover, Lp(a) regulates endothelial cell synthesis of plasminogen activator inhibitor-1 (PAI-1), a major fibrinolytic protein [11,29].

### 1.4. RNA-Based Therapy

Nucleic acid-based therapies allow for selective gene silencing, thus preventing the production of proteins that may cause or enhance disease. RNA therapies include antisense oligonucleotides (ASOs), small interfering RNAs (siRNAs), microRNAs (miRNAs), RNA aptamers, and mRNAs [30,31].

RNA-based therapeutics are currently being developed for the prevention and management of cardiovascular diseases (CVD). Antisense oligonucleotides and short-interfering RNA therapeutics to lower Lp(a) have been initiated.

ASOs are short, synthetic single-stranded strings of nucleotides designed to bind to the RNA target with complementary sequences through Watson-Crick base paring (hybridization) [32]. After subcutaneous injections, ASO enters the cell and nucleus and then binds to the mRNA target, leading to RNAse–H activation and subsequent mRNA degradation. Hybridization of ASO with RNA, which encodes the targeted protein, prevents its expression [33].

The antisense strand is resistant to cleavage and can bind to another target mRNA, leading to the long half-lives of these drugs (3–4 weeks) [34,35].

siRNAs are synthetic, double-stranded RNAs, are typically at least twice the size of ASOs, and contain a more negative charge. siRNA consists of sense (passenger) and antisense (guide) strands, which are separated when they enter the cytoplasm [36]. Then the siRNA molecule is incorporated into an RNA-induced silencing complex (RISC), a nuclease-containing multi-protein complex [37]. The activated RISC consists of the guide strand that binds precisely to the target sequence of mRNA. Argonaute-2 cleaves the mRNA leading to its degradation, thereby reducing protein levels [38]. siRNAs remain bound to the RISC, targeting multiple RNA copies, thereby having a long-term effect [39].

Both ASO and siRNA can’t be used as drugs without chemical modifications, which are required to improve their pharmacokinetics and pharmacodynamics [31].

Moreover, early generations of antisense oligonucleotides, as well as small interfering RNA therapies, had safety issues. Drug-induced thrombocytopenia was associated with the use of several ASOs, while peripheral neuropathy and excessive mortality were reported after treatment with GalNAc–siRNA revusiran [40,41,42]. The next generation of RNA therapeutics has several chemical modifications implemented to overcome these issues and improve the safety profile [43].

ASO modifications mainly include chemically modifying some nucleotides, altering the backbone, changing the sugars, or acting on the phosphodiester linkage. These modifications not only protect from premature degradation of a drug but also improve the safety profile by reducing the risk of activating an innate immune response leading to adverse effects related to systemic inflammation [44]. The most common modifications used in ASO are 2′ ribose modifications, which include 2′-O-methyl (2′-OMe), 2′-fluoro (2′-F), 2′-O-methoxyethyl (2′-MOE), 2′,4′-constrained 2′-O-ethyl and locked nucleic [35]. Most of these 2′ modifications can also be adapted to siRNAs.

The activity of ASOs and siRNAs rely on reaching the interior of the target cells. The most successful and widely used delivery method is a conjugation of siRNA or ASO molecule to triantennary N-acetylgalactosamine (GalNAc). GalNAc is a ligand for asialoglycoprotein receptors, which are predominantly expressed on liver hepatocytes [45,46,47]. The liver plays an important role in lipid metabolism, thus GalNAc-conjugated siRNAs and ASOs have tremendous potential for treating dyslipidemia.

## 2. Lp(a) Lowering Drugs under Clinical Development

Currently under clinical trials are four Lp(a)-reducing RNAi therapeutics, three of them, olpasiran, SLN360, and LY3819469, are using the siRNA principle, and one, pelacarsen, is an ASO (Table 4 and Figure 2).

### 2.1. Pelacarsen

Pelacarsen (TQJ230; formerly IONIS-APO(a)-LRX, AKCEA-APO(a)-LRX, ISIS 681257) is a 2′-MOE chimeric second-generation antisense oligonucleotide covalently bonded to GalNAc, which ensures uptake by hepatocytes through the asialoglycoprotein receptor (Table 4). Pelacarsen enters the cell and binds to the mRNA in the nucleus by Watson and Crick base pairing, leading to RNAse–H activation and subsequent mRNA degradation, thus preventing the production of the apo(a) protein (Figure 2) [49]. Pelacarsen is administered subcutaneously once every month.

The first version of this drug without the GalNAc ligand, called APO(a)-Rx, was tested in phase 1 and phase 2 (NCT02160899) trials that indicated substantial decreases in Lp(a) and good tolerability [50,51].

APO(a)-Rx was chemically modified and conjugated with GalNAc, which provides liver targeting, this version was called APO(a)-LRx (pelacarsen) [52]. It was found that APO(a)-LRx was 30 times more potent than APO(a)-Rx and a 10 times lower dose caused the same reduction of Lp(a) level with enhanced tolerability [51].

Several clinical trials evaluating the efficacy and safety of pelacarsen have been completed, but not all results have been published (Table 5).

Clinical trial phase 1/2a (NCT02414594) enrolled 58 healthy participants who had Lp(a) >75 nmol/L. Patients were divided into two cohorts: 28 patients received a single dose of 10, 20, 40, 80, and 120 mg of pelacarsen or placebo, and 30 subjects received multiple doses of 10, 20, or 40 mg of the therapeutics at day 1, 3, 5, 8, 15, and 22 or placebo. Significant dose-dependent reductions in Lp(a) concentration were noted in both cohorts. The biggest changes (85% and −79%) were observed at 120 mg and 80 mg of a single dose, respectively, and −92% at 40 mg of multiple doses. The effects of therapy with pelacarsen were sustained at day 90, with significant Lp(a) reductions of 46% and 44% at the single dose of 80 mg and 120 mg, respectively, while effects of multiple dosing of 10, 20, and 40 mg were sustained at 113 days after the last dose, with Lp(a) reductions of 39, 53, and 58%, respectively [51]. In this study, no treatment-related serious adverse events (SAE), injection-site reactions, or influenza-like symptoms were observed. Additionally, it was indicated that pelacarsen decreased LDL-C, apolipoprotein B (apo B), and oxidized phospholipids on apo B and apo(a) [51]. However, it should be noted that the current method of measurement of LDL-C also includes the level of Lp(a) cholesterol; thus, a lower level of LDL-C might be associated with Lp(a) reduction [53].

In clinical trial phase 2 (NCT03070782), 286 enrolled patients with established CVD and Lp(a) > 60 mg/dL received pelacarsen (20, 40, or 60 mg Q4W; 20 mg Q2W; or 20 mg QW), or placebo for 6 to 12 months. Pelacarsen administration caused a dose-dependent reduction in the level of Lp(a). The biggest changes, −72% and −80%, were noted at a dose of 60 mg every 4 weeks and at 20 mg every week, respectively [54]. In a group with a dose of 20 mg QW, 98% of patients achieved an Lp(a) level of 50 mg/dL or lower. The Lp(a) returned to the baseline value within 16 weeks after the last dose [54]. Treatment with pelacarsen was found to be well tolerated. The most frequently reported adverse events (AE) were injection site reactions, which occurred in 27% of patient groups with pelacarsen administration and 6% of participants in the placebo group. Other observed AEs were urinary tract infections (13% drug versus 6% placebo), myalgia (12% vs. 11%), headache (11% vs. 8%), and influenza-like symptoms (7% vs. 6%). Most of the reported AEs were mild or moderate [54]. No significant effect on liver function, kidney function, or platelet count was observed [54]. As indicated in previous trials, it was noted that pelacarsen decreased LDL-C, apolipoprotein B (apo B), and oxidized phospholipids on apo B and apo(a) [51,54].

Data from four clinical trials with pelacarsen indicated that the reduction of Lp(a) was independent of LPA genetic variants and isoform size [55].

Currently, pelacarsen is under clinical trial phase 3 Lp(a) HORIZON (Table 5). Lp(a) HORIZON trial (NCT04023552) (https://clinicaltrials.gov/ct2/show/study/NCT04023552; accessed on 29 October 2022) started on December 2019, and the number of enrolled participants is 8323. The key inclusion criteria are Lp(a) ≥ 70 mg/dL at the screening visit, an optimal LDL cholesterol-lowering treatment, an optimal treatment of other cardiovascular risk factors, and a myocardial infarction or an ischemic stroke ≥3 months from screening, and randomization to ≤10 years prior to the screening visit or a clinically significant symptomatic peripheral artery disease. Patients are injected monthly with 80 mg of pelacarsen or placebo subcutaneously for 4–5 years. The estimated study completion date is May 2025. The results of this study may provide an answer to whether a marked decrease in Lp(a) level translates into a clinical benefit in terms of cardiovascular outcomes.

### 2.2. Olpasiran

Olpasiran (formerly AMG-890, ARO-LPA) is an siRNA conjugated with GalNac designed to directly inhibit LPA messenger RNA translation in hepatocytes (Table 4) [41]. Degradation of LPA mRNA prevents its translation into apo(a), which leads to reducing the synthesis of Lp(a) (Figure 2). The macromolecule of olpasiran is modified with 2′-fluoro and 2′-methoxy substitutions together with phosphorothioate internucleotide linkages at the termini to enhance stability [48,56]. The GalNac moiety protects the macromolecule of siRNA from degradation and enhances cellular uptake [57]. Olpasiran is administered subcutaneously once every 3 or 6 months.

Olpasiran was found as the best candidate for clinical testing among a group of 108 siRNA sequences specific for human LPA determined using a bioinformatics algorithm [54]. Studies on olpasiran pharmacodynamics in transgenic mice indicated more than 80% decrease in serum Lp(a) concentration after a single dose of 1 m/kg of olpasiran. The reduction of Lp(a) values was maintained for 5 weeks after dosing [54]. Studies on cynomolgus monkeys indicated more than an 80% reduction in serum Lp(a) concentration after dosing of 3 mg/kg and 10 mg/kg of olpasiran [58].

Results from three clinical trials (NCT03626662, NCT04987320, NCT04270760) with olpasiran are published and several trials are underway (Table 5).

Phase 1 clinical trial NCT03626662 started on July 2018, and the estimated study completion date is April 2023 (https://clinicaltrials.gov/ct2/show/NCT03626662; accessed on 29 October 2022). The value of Lp(a) concertation at the screening of enrolled subjects was between 70 and 199 nM (cohorts 1–5) and ≥200 nM (cohorts 6 and 7). Sixty-four participants received a single dose of olpasiran or a placebo. Olpasiran were well tolerated, and only one patient experienced an injection site reaction. The most common adverse events were headache (10% drug; 25% placebo) and upper respiratory tract infection (15% drug; 13% placebo). No serious adverse events were reported [58]. A dose-responsive manner of Lp(a) reduction was found. In cohorts 1–5 (3, 9, 30, 75 or 225 mg) the mean percent change from baseline was −71% to 97%, and in cohorts 6 (9 mg) and 7 (75 mg) was −198 nM to −266 nM [54]. The biggest reduction of Lp(a) level from baseline was observed between days 43 and 71, then the values of Lp(a) started to increase but remained well below the placebo levels at day 225 [58].

Sohn et al. [59] evaluated the pharmacokinetics, pharmacodynamics, and tolerability of a single dose of olpasiran in healthy Japanese, and non-Japanese participants in phase I open-label study (NCT04987320) (https://clinicaltrials.gov/ct2/show/NCT04987320; accessed on 29 October 2022) (Table 5). A total of 27 participants were enrolled. Japanese participants received a single dose of 3, 9, 75, or 225 mg of olpasiran, and non-Japanese participants received a single dose of 75 mg. Lp(a) was decreased in a dose-dependent manner and a greater effect of reduction was achieved with higher doses. Mean percentage changes from baseline in Lp(a) ranged from –56% to –99%, with maximal reduction noted at day 57. A reduction in Lp(a) started from day 4 and returned to within 50% of baseline in the 3 and 9 mg dose cohorts by day 225. Lp(a) reductions were similar between the Japanese and non-Japanese groups. Olpasiran was well tolerated, adverse events were mild in severity, and no serious or fatal AEs were reported [59].

OCEAN(a)-DOSE was a dose-finding clinical trial phase 2 (NCT04270760) (https://clinicaltrials.gov/ct2/show/NCT04270760; accessed on 29 October 2022) aimed at evaluating the efficacy, safety, and tolerability of olpasiran. The key inclusion criteria were lipoprotein (a) > 150 nmol/L and evidence of atherosclerotic CVD [60]. A total number of 281 patients were enrolled. The median concentration of Lp(a) and LDL-C at baseline was 260.3 nmol/L and 67.5 mg/dL, respectively. Participants were randomly allocated to one of four treatment groups with different doses of olpasiran (10, 75, or 225 mg Q12W, or 225 mg of Q24W) or matched placebo [61]. O’Donoghue et al. [61] reported that after 36 days of olpasiran therapy, Lp(a) concentration was decreased in a dose-dependent manner, by −70.5% at 10 mg dose, −97.4% at a 75 mg dose, and −101.1% at a 225 mg dose Q12W, and −100.5% at a 225 mg dose of Q24W (*p* < 0.001 for all comparisons with baseline). Almost all participants treated with olpasiran achieved a lipoprotein (a) concentration lower than 125 nmol/L [60]. The overall incidence of adverse events, serious adverse events, and adverse events leading to discontinuation were similar among patients who received olpasiran and those who received a placebo. The most common AEs were injection-site reaction (17% vs. 11% placebo), primarily pain, and hypersensitivity reaction (6% vs. 2% placebo). The OCEAN(a)-dose was a short-term trial. Longer and larger randomized clinical trials are necessary to evaluate the long-term efficacy and safety of olpasiran, as well as the effect of olpasiran therapy on cardiovascular disease [61].

Currently, olpasiran is under a planned phase 3 clinical trial (NCT05581303) (https://clinicaltrials.gov/ct2/show/NCT05581303; accessed on 29 October 2022) (Table 5).

Study NCT05581303 will start on December 2022, and the estimated completion date is December 2023. The target for enrolment is 6000 participants with Lp(a) ≥ 200 nmol/L during screening, and history of ASCVD. The results of these two studies will provide new insights into the pharmacokinetic and pharmacodynamic properties of olpasiran, the best dosing, and its impact on cardiovascular events.

Moreover, olpasiran is under two phase 1 clinical trials with patients with mild, moderate, or severe hepatic impairment compared to participants with normal hepatic function (NCT05481411) and subjects with normal renal function, and participants with various degrees of renal impairment (NCT05489614). Both studies started in September 2022 and are currently recruiting participants. The estimated study completion date is August 2023 (Table 5).

### 2.3. SLN360

SLN360 is an siRNA conjugated to GalNAc3 that interferes with the biosynthesis of Lp(a) in the liver [62]. The target of SLN360 is the transcription product of the LPA gene—LPA mRNA (Table 4). SLN360 forms a complex with LPA mRNA, leading to its degradation, as a result of which translation cannot take place, nor the production of apo (a) necessary for the production of Lp(a) (Figure 2) [62]. SLN360 is a drug administered by subcutaneous injection.

In an in vitro and in vivo study by Rider et al., the efficacy of SLN360 in reducing Lp(a) levels was assessed. In an in vitro study using human hepatocytes, SLN360 was found to reduce the amount of LPA mRNA. In an in vivo study using *cynomolgus* monkeys, it was found that SLN360 caused up to 95% reduction in serum Lp(a) concentration and, importantly, this effect was long-lasting [63]. Toxicological analyzes, both in vitro and in vivo, showed satisfactory safety of using SLN360 [64].

A phase I randomized clinical trial, APOLLO, by Nissen et al., involved 32 adults with baseline plasma Lp(a) concentrations ≥150 nmol/L without known CVD. Subjects were randomized to a single subcutaneous dose of SLN360 of 30, 100, 300, or 600 mg vs. placebo, and were followed for 150 days. It was shown that the decrease in plasma Lp(a) concentration was dependent on the dose of SLN360 used. The greatest reduction in the plasma Lp(a) concentration, by as much as −98%, was observed in subjects who received SLN360 at a dose of 600 mg (placebo: −10% (95% CI: −16 to 1); 30 mg: −46% (95% CI: −64 to −40); 100 mg: −86% (95% CI: −92 to −82); 300 mg: −96% (95% CI: −98 to −89); 600 mg: −98% (95% CI: −98 to −97)). At 150 days after injection of SLN360, the plasma Lp(a) concentration was still clinically significantly reduced (SLN360 300 mg: −70%; SLN360 600 mg: −81%). SLN360 had a satisfactory safety profile [65].

Currently, two clinical trials are underway using SLN360 (NCT05537571 and NCT04606602) (Table 5). The first, randomized and placebo-controlled phase I trial is to target 88 subjects and assess the safety, tolerability, pharmacokinetics, and pharmacodynamics of SLN360 in patients with elevated Lp(a) concentrations. The study is scheduled for completion in November 2022. A second, randomized and placebo-controlled phase II study is to target 160 subjects and assess the effectiveness of SLN360 in patients with elevated Lp(a) concentrations and high cardiovascular risk. The study is scheduled for completion in November 2024.

### 2.4. LY3819469

The newest therapeutic candidate for the Lp(a) lowering is a GalNAc conjugated mixed 2′-o-me, 2′-fluoro and unmodified Dicer siRNA (Table 4) (Figure 2) [66] called LY3819469. Currently, LY3819469 is under clinical trial with phase 1 (NCT04914546). Clinical trial NCT04914546 started on 14 June 2021, and the target number of participants is 66. The main purpose of the study is to evaluate the safety, tolerability, pharmacokinetics, and pharmacodynamics of the LY3819469 in healthy participants with high Lp(a) concentrations. The estimated study completion date is November 2022 (https://clinicaltrials.gov/ct2/show/NCT04914546; accessed on 29 October 2022) (Table 5).

**Table 5 pharmaceuticals-15-01573-t005:** Summary of the results of clinical trials with emerging Lp(a) lowering therapeutics.

Drug	Clinical Phase/ Status	NCT Number/ Reference	Population/ Sample Size	Duration	Dose/ Treatment Arms	Key Results	Kay Safety Data
**Pelacarsen**	Phase 1/2a completed Viney et al., 2016 [51]	NCT02414594	Healthy Lp(a) ≥ 75 nmol/L (30 mg/dL); *n* = 58	22 days	Single ascending dose: 10, 20, 40, 80, 120 mg; Multiple ascending doses: 10, 20, 40 mg in 6 doses each	Lp(a) reduction: Single dose −26% to −85%; Multiple doses −66% to −92%	No SAE
	Phase 2 completed Tsimikas et al., 2020 [50]	NCT03070782	Lp(a) ≥ 60 mg/dL and CVD; *n*= 286	6–12 months	20 mg Q1W; 20 mg Q2W; 20, 40, 60 mg Q4W	Lp(a) reduction: −35% to −80%	Injection site reactions (27% drug vs. 6% placebo), urinary tract infection (13% drug vs. 6% placebo), myalgia (12% vs. 11%), headache (11% vs. 8%), influenza-like symptoms (7% vs. 6%)
	Phase 3 ongoing	NCT04023552	Lp(a) ≥ 70 mg/dL and CVD; *n* = 8323	4–5 years	80 mg Q4W	NA	NA
	Phase 1 completed	NCT05337878	L(a) ≥ 15 nmol/L (8 mg/dL); healthy Japanese *n* = 29	204 days	Single ascending dose: 20, 40, 80 mg; Multiple ascending doses: 80 mg Q4W	NA	NA
	Phase 3 recruiting	NCT05305664	Lp(a) > 60 mg/dL and CVD; *n* = 60 (target)	~208 days	80 mg Q4W vs. placebo	NA	NA
	Phase 1 recruiting	NCT05026996	Subjects with mild hepatic impairment compared to matched healthy participants *n* = 16	60 days	Single dose	NA	NA
**Olpasiran**	Phase 1 ongoing	NCT03626662 Koren et al., 2022 [58]	Lp(a) ≥ 70 nmol/L ≤ 199 nmol/L, *n* = 40; Lp(a) ≥ 200 nmol/L, *n* = 24	225 days	Single dose; 3 mg, 9 mg, 30 mg, 75 mg, 225 mg	Lp(a) reduction: −71 to −97%	Headache (10% drug vs. 25% placebo), upper respiratory tract infection (15% drug vs. 13% placebo), injection site reaction (one patient); no SAE
	Phase 1 completed	NCT04987320 Sohn et al., 2022 [59]	Lp(a) ≥ 70 nmol/L (≥27 mg/dL); healthy Japanese and non-Japanese patients; *n* = 27	225 days	Single dose: 3 mg, 9 mg, 75 mg, 225 mg	Lp(a) reduction: –56% to –99%	Headache (one patient−16.7%), vitreous floaters (one patient 16.7%); no SAE
	Phase 2 ongoing	NCT04270760 O’Donghaou et al., 2022 [60,61]	L(a) > 150 nmol/L and ASCVD; *n* = 281	336 days	10 mg Q12W, 75 mg Q12W, 225 mg Q12W, 225 mg Q24W, placebo	Lp(a) reduction: −70.5% to −100.5%	The most common AE: injection-site reaction (17% vs. 11% placebo) hypersensitivity reaction (6% vs. 2% placebo)
	Phase 3 not yet recruiting	NCT05581303	Lp(a) ≥ 200 nmol/L and ASCVD *n* = 6000 (target)	4 years	Q12W vs. placebo	NA	NA
	Phase 1 recruiting	NCT05481411	Subjects with mild, moderate, or severe hepatic impairment compared to participants with normal hepatic function *n* = 24 (target)	85 days	Single dose on day 1	NA	NA
	Phase 1 recruiting	NCT05489614	Subjects with normal renal function and participants with various degrees of renal impairment *n* = 32 (target)	85 days	Single dose on day 1	NA	NA
**SLN360**	Phase 1 completed	NCT04606602 Nissen et al., 2022 [65]	Lp(a) ≥ 150 nmol/L; *n* = 32	150 days	Single dose: 30; 100; 300 or 600 mg vs. placebo	Lp(a) reduction −46% to −98%	AEs were generally mild, most commonly low-grade injection site events (grades 1 and 2) and headache; no SAE
	Phase 2 not yet recruiting	NCT05537571	Lp(a) > 125 nmol/L and high risk of ASCVD event; *n* = 160 (target)	240 days	Three different doses vs. placebo	NA	NA
	Phase 1 Recruiting	NCT04606602	L(a) ≥ 125 nmol/L *n* = 88 (target)	201 days	Single or multiple doses vs. placebo	NA	NA
**LY3819469**	Phase 1 ongoing	NCT04914546	Part A: Healthy, high Lp(a) levels Part B Japanese participant; *n* = 66 (target)	part A: 53 weeks Part B: 29 weeks	NA	NA	NA

Lp(a)—lipoprotein (a); SAE—serious adverse event; QXW—once every X weeks; CVD—cardiovascular disease; NA—, no data are available; ASCVD—atherosclerotic cardiovascular disease; AEs—adverse event.

## 3. Potential Risk Associated with Low Levels of Lp(a)

Many studies have indicated that a very low level of Lp(a), approximately below 7 mg/dL, is associated with an increased risk of type 2 diabetes mellitus (TD2M) [67,68,69,70]. However, the mechanism of this association is unknown [71,72]. Additionally, the casualty of this relationship has not been revealed [69,73]; thus, the question remains whether new therapies to lower Lp(a) may increase the risk of developing T2DM.

Taking into consideration that patients with high levels of Lp(a) have a risk of CVD, and very low levels reveal an increased risk of developing T2DM, it might be safe to lower Lp(a) to medium levels.

Patients at risk of CVD have Lp(a) above 70 mg/dL, so a reduction of up to 80% theoretically will not be associated with the development of T2DM, as this association has been observed in patients with an Lp(a) level below 7 mg/dL. Moreover, cardiovascular benefits of Lp(a)-lowering are likely to outweigh any potential risks related to T2DM [47].

## 4. ASO versus siRNA in Lowering Lp(a)

Taking into consideration the advancement of clinical trials, pelacarsen comes first, second is olpasiran then SLN360 and LY3819469 takes the last place, which is currently under a phase 1 clinical trial. Pelacarsen, olpasiran, and SLN360 cause reductions in Lp(a) levels that are likely to be clinically significant. siRNA molecules showed greater potency than ASO, olpasiran, and SLN360 caused Lp(a) reduction of about 95% and 98%, respectively, in phase 1 trials [58] compared with 80% reduction with pelacarsen in a phase 2 study [54].

In terms of dosing frequency and duration of the Lp(a) lowering effect, siRNA drugs have had better results than ASO. ASOs are mainly administered every month, while the longer duration of action of siRNA therapies allows for a more extended period between dosing [74,75,76].

A single dose of 75 mg and 225 mg of olpasiran caused about 95% reduction of Lp(a) for several months [58], while pelacarsen caused 72% and 80% reduction of Lp(a) in a similar period of time at a dose of 60 mg administered monthly, or 20 mg administered weekly, respectively [54]. A less frequent administration schedule offers a potential advantage of siRNA compared to ASO as this certainly has a greater benefit for the patient.

Injection site reactions were the most common adverse event reported after pelacarsen administration in a phase 2 study, while this AE was less frequent in phase 1 and phase 2 studies with olpasiran [54,58,61].

If siRNA drugs, such as olpasiran, SLN360, and LY3819469 are demonstrated to be safe and effective in the ongoing and planned clinical trials, they stand to offer options for CVD risk modification in patients with elevated Lp(a).

Treatment duration was relatively short in most of the currently completed clinical trials, ipso facto the long-term effects of ASOs and siRNAs are unknown.

## 5. Conclusions

Lp(a) is established as a risk factor for ASCVD and aortic valve disease. Currently, there is no available therapy dedicated to lowering Lp(a). The development of olpasiran, SLN360, and pelacarsen is still ongoing; however, available results of conducted clinical trials have shown promise in achieving a safe and effective therapy for the reduction of Lp(a) level. Clinical trials with pelacarsen are the most advanced among RNA-based Lp(a)-lowering drugs. A Phase 3 study with pelacarsen, which is planned to finish in May 2025, may provide an answer to whether or not a marked reduction of Lp(a) brings clinical benefit in terms of cardiovascular events.

Two siRNA drugs, olpasiran and SLN360, are full of promise to achieve even greater Lp(a) reduction than pelacarsen, with less frequent dosing. Moreover, there is a new siRNA candidate for an Lp(a)-lowering drug named LY3819469, which is under a phase 1 study. Further research is needed to introduce those new therapeutics into clinical practice.

## Figures and Tables

**Figure 1 pharmaceuticals-15-01573-f001:**
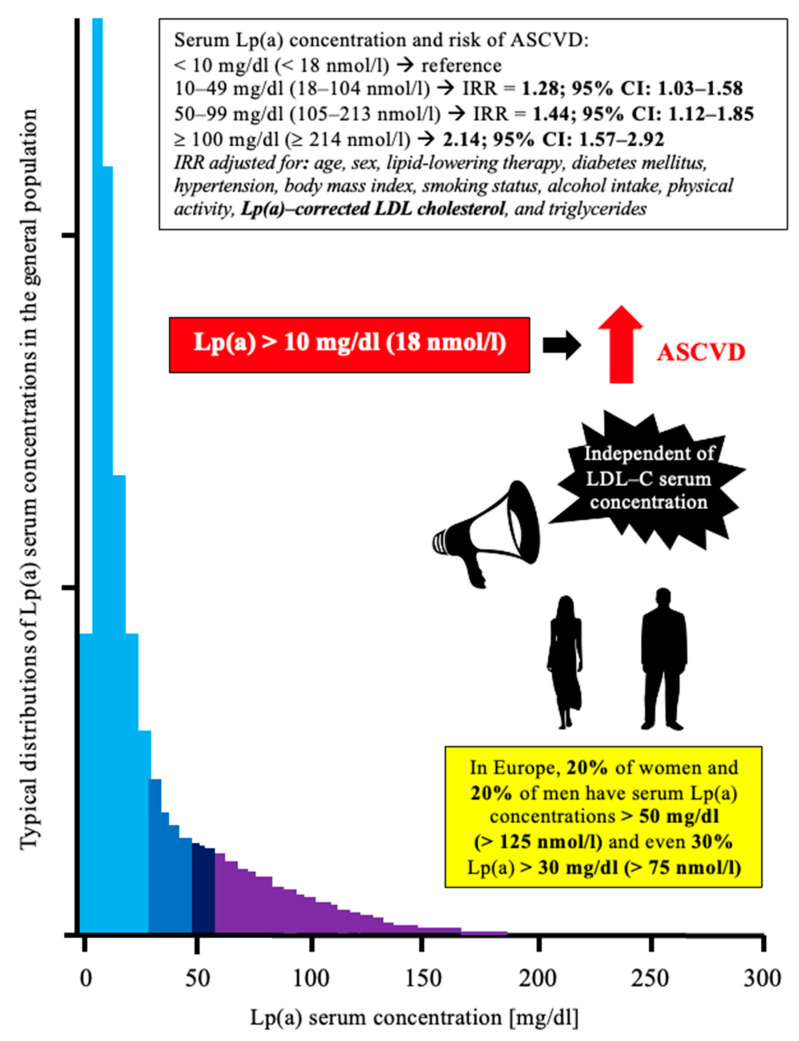
Distribution of serum Lp(a) concentrations in the general population and its relationship to the risk of ASCVD. Prepared on the basis of [11,12,13]. ASCVD—atherosclerotic cardiovascular disease; LDL-C—low density lipoprotein cholesterol; Lp(a)—lipoprotein (a).

**Figure 2 pharmaceuticals-15-01573-f002:**
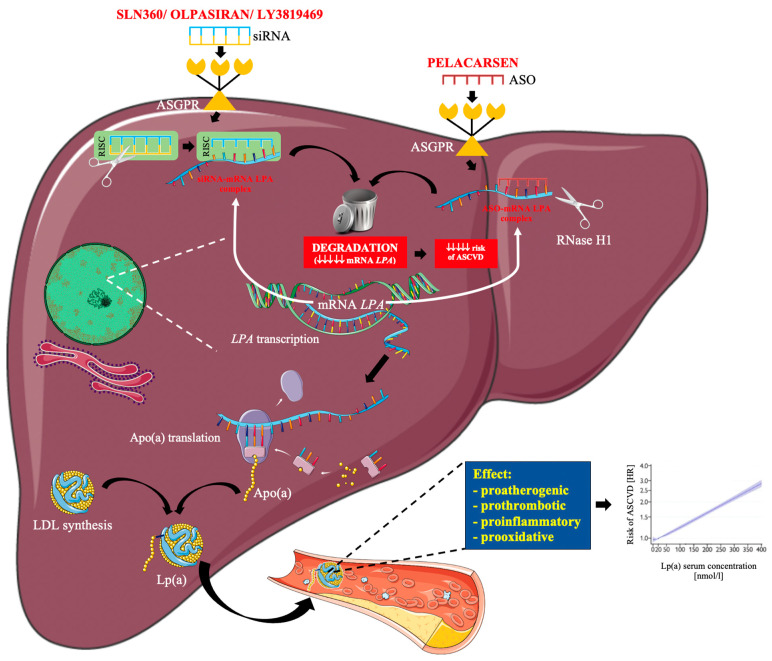
Mechanism of action of SLN360, olpasiran, LY3819469 and pelacarsen [11,15,48]. ASGPR—asialoglycoprotein receptor; ASO—antisense oligonucleotides; siRNA—small interfering RNA; LPA—lipoprotein (a) gene; LDL—low density lipoprotein; ASCVD—atherosclerotic cardiovascular disease; Lp(a)—lipoprotein (a); apo (a)—apolipoprotein (a); RICS—RNA-induced silencing complex. The following was used in the preparation of the figure: https://smart.servier.com (accessed on 29 October 2022; free-access).

**Table 1 pharmaceuticals-15-01573-t001:** Lp(a) serum concentrations and cardiovascular risk.

Author, Year, [Ref]	Characteristics of the Study Group	Key Results
**Lp(a) and ASCVD Risk**		
Rikhi R. et al., 2022, [9]	4585 subjects from Multi-Ethnic Study of Atherosclerosis (MESA), follow-up: 13.4 years	Elevated Lp(a), regardless of baseline LDL-C, significant increased ASCVD risk
Afshar M. et al., 2020, [16]	2606 subjects from Framingham Offspring	Regardless of LDL-C, were associated with an increased risk of ASCVD
Kaiser Y. et al., 2022, [17]	191 patients with ASCVD	Increased Lp(a) are associated with accelerated progression of low-attenuation plaque in patients with advanced multivessel CAD despite receiving guideline- based preventative therapies
Nurmohamed N. et al., 2022, [18]	12,437 subjects with ASCVD	Patients with Lp(a) > 99th percentile had an OR of 2.64 for ASCVD [95% CI: 1.45–4.89] and 3.39 for MI (95% CI: 1.56–7.94). Importantly, the addition of Lp(a) to ASCVD risk algorithms led to 31% and 63% being reclassified into a higher risk category for SCORE and Second Manifestations of ARTerial disease (SMART), respectively
Willeit P. et al., 2014 [19]	826 subjects	Increased Lp(a) predict 15-year ASCVD outcomes and improves ASCVD risk prediction, particularly in intermediate-risk groups
Kampstrup P. et al., 2013, [20]	8720 subjects	Extreme Lp(a) level substantially improved MI and CAD risk prediction
**Lowering of Lp(a) and the risk of ASCVD**		
O’Donoghue M. et al., 2019, [21]	25,069 patients with stable CAD treated with evolocumab; follow-up: 2.2 years	Significant correlation between a 15% reduction in the risk of major coronary events (95% Cl: 2–25%; *p* = 0.0199) and a reduction in serum Lp(a) concentration by 25 nmol/L after adjustment for LDL-C
Bittner V. et al., 2020, [22] and Szarek M. et al., 2020, [23]	ODYSSEY OUTCOMES study in 18,924 ACS patients; follow-up: 2.8 years	A 5 mg/dL reduction in Lp(a) was associated with a significant 2.5% reduction in cardiovascular events
Burgess S. et al., 2018, [24]	62,240 patients CAD compared with a control group of 127,000 subjects	Each 10 mg/dL reduction in Lp(a) was associated with a 5.8% reduction in the risk of CAD (OR = 0.94; 95% Cl: 0.93–0.95)

Lp(a)—lipoprotein (a); LDL-C—low-density lipoprotein cholesterol; ASCVD—atherosclerotic cardiovascular disease; CAD—coronary artery disease; OR—odds ratio; MI—myocardial infarction; 95% CI—95% confidence interval; ACS—acute coronary syndrome.

**Table 2 pharmaceuticals-15-01573-t002:** Effect of lipid-lowering drugs on Lp(a) serum concentration [5,25,26].

Lipid-Lowering Drug	Effect on Lp(a) Serum Concentration
Niacin	Reduction; 30%
Statins	Possible increase; 6–10%
Ezetimibe	Possible reduction; 0–7%
Bempedoic acid	No effect
Fibrates	Minimal, possible increase in setting of HTG
Bile acid sequestrants	No effect
PCSK9 inhibitors	Reduction; 20–30%
Inclisiran	Reduction; 15–26%
Mipomersen	Reduction; 25%
CETP inhibitors	Reduction; 25%
ASO based drugs	Reduction; 70–90%
siRNA based drugs	Reduction 70–98%
Lipoprotein apheresis	Reduction; 20–30%
Lipoprotein (a) apheresis	Reduction; 70–80%

HTG—hypertriglyceridemia; PCSK9—proprotein convertase subtilisin/kexin type 9; CETP—cholesteryl ester transfer protein; ASO—antisense oligonucleotides; siRNA—small interfering RNA.

**Table 3 pharmaceuticals-15-01573-t003:** Recommendations regarding the Lp(a) measurement [5].

Recommendations	Class	Level
Lp(a) concentration should be measured at least once in every adult individual’s life.	**IIa**	**C**
Measurement of Lp(a) should be considered in all patients with premature onset of cardiovascular disease, the lack of expected statin therapy effect, and in those with a borderline risk between moderate and high, for better risk stratification.	**IIa**	**C**
Measurement of Lp(a) may be considered in patients with very high cardiovascular risk and atherosclerotic cardiovascular disease, in patients with familial hypercholesterolaemia, and in pregnant women as a prevention of pre-eclampsia or miscarriage, in recurrent pregnancy loss, or intrauterine growth restriction.	**IIb**	**C**

**Table 4 pharmaceuticals-15-01573-t004:** RNA-based Lp(a)-lowering therapeutics under clinical development.

Drug	Type	Chemical Modification	Delivery System	Targeting Gene	Company
Pelacarsen (TQJ230; formerly IONIS-APO(a)-LRX, AKCEA-APO(a)-LRX, ISIS 681257)	ASO	2′-O-MOE	GalNAc	LPA	Novartis Pharmaceuticals
Olpasiran (AMG-890, ARO-LPA)	siRNA	2′-O-Me	GalNAc	LPA	Amgen, Arrowhead Pharmaceuticals
SLN360	siRNA	2′-O-Me	GalNAc	LPA	Silence Therapeutics plc
LY3819469	siRNA	2′-O-Me; 2′-F	GalNAc	LPA	Eli Lilly and Company

ASO—antisense oligonucleotides; siRNA—small interfering RNA; LPA—lipoprotein (a) gene; 2′-F, 2′-fluoro; 2′-O-Me, 2′-methoxy; 2′-O-MOE, 2′-methoxyethyl.

## Data Availability

Not applicable.

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
