# Peer review of "Targeted Treatment against Lipoprotein (a): The Coming Breakthrough in Lipid Lowering Therapy"

_pharmaceuticals, 2022, doi:10.3390/ph15121573_

Round 1

Reviewer 1 Report

In their review article titled ` Targeted Treatment Against Lipoprotein (a) - the Coming Breakthrough in Lipid Lowering Therapy`, Sosnowska et al. have presented an comprehensive updated overview of the clinical significance of Lp(a)] as a risk factor for ASCVD, and the promising therapeutic potential of drugs based on RNA technology, such as pelacarsen, olpasiran, SLN360 and LY3819469, which are currently being evaluated in several clinical trials. The authors have done a commendable job to draft such a timely relevant review. The review also includes schematic presentations that effectively summarize the highlights of the discussed topics. I only have a couple of minor suggestions to make which in my opinion will further enrich the review.

1.  I request the authors to include a tabulation information of the clinical studies discussed under the heading Lipoprotein (a) – brief overview for clinicians.

2.    Please elaborate the molecular mechanisms driving Lipoprotein (a) atherothrombotic complications under a separate heading. The present information is too brief.

Author Response

Reviewer 1

In their review article titled ` Targeted Treatment Against Lipoprotein (a) - the Coming Breakthrough in Lipid Lowering Therapy`, Sosnowska et al. have presented an comprehensive updated overview of the clinical significance of Lp(a)] as a risk factor for ASCVD, and the promising therapeutic potential of drugs based on RNA technology, such as pelacarsen, olpasiran, SLN360 and LY3819469, which are currently being evaluated in several clinical trials. The authors have done a commendable job to draft such a timely relevant review. The review also includes schematic presentations that effectively summarize the highlights of the discussed topics. I only have a couple of minor suggestions to make which in my opinion will further enrich the review. 

AUTHOR RESPONSE:

Dear Reviewer, thank you very much for your positive opinion about our work - it is very important to us. We have implemented all your tips.

Below are the answers to individual comments.

  1. I request the authors to include a tabulation information of the clinical studies discussed under the heading Lipoprotein (a) – brief overview for clinicians.

AUTHOR RESPONSE:

We have summarized the most important results of the studies described in this subchapter in Table 2.

  1.   Please elaborate the molecular mechanisms driving Lipoprotein (a) atherothrombotic complications under a separate heading. The present information is too brief.

AUTHOR RESPONSE:

We have summarized the pathophysiological mechanisms in a separate subsection. We've expanded it a bit. We have summarized the most important things.

Reviewer 2 Report

Dear Authors:

Please pay attention to the following points:

1-    What is the head topic of the first group of subheadings? Background?

2-    What is the source of Fig 1? Is it an original finding?

3-    If the tables and figures are adopted from other studies, please take note about the copyright

4-    What methods did you use to select relevant studies? Can you explain about how you gathered the information?

Author Response

Reviewer 2

  1. What is the head topic of the first group of subheadings? Background?

AUTHOR RESPONSE:

Thank you for your suggestion. We added the head topic of the first 3 subheadings and changed the headlines 5-8 as subtitles of heading 2.

  1. What is the source of Fig 1? Is it an original finding?

AUTHOR RESPONSE:

Figure 1 is original, prepared on the basis of [11-13].

  1. If the tables and figures are adopted from other studies, please take note about the copyright

AUTHOR RESPONSE:

            All tables and figures are original.

  1. What methods did you use to select relevant studies? Can you explain about how you gathered the information?

AUTHOR RESPONSE:

We searched PubMed database up to November 2022 to find originals and review studies on lipoprotein (a) lowering therapy, pelacarsen, olpasiran, SLN360. For the search strategy, the following keywords were used: lipoprotein (a), lowering lipoprotein (a), RNA therapy, pelacarsen, olpasiran, SLN360. Additionally, we use: https://clinicaltrials.gov/ to find conducted and ongoing clinical trials with pelacarsen, olpasiran, SLN360 and LY3819469.